# A study of the influence of sports venues on the intra-city population layout based on multi-source data—Taking Xi'an city and Zhengzhou city as examples

**Shulin Zhang**[1☯], **Xuejie Zhang**[2☯], **Yang Liu**[1]*

**1** Physical Education College of Xinjiang Normal University, Urumqi, China, **2** College of Geography and Environment, Shandong Normal University, Ji'nan, China

☯ These authors contributed equally to this work.
* xj_liuyang1020@163.com

## Abstract

Revealing the influence of sports sports venues on the population in the built-up areas of cities contributes to the high-quality development of cities and the well-being of people. This study applies kernel density estimation to characterize the distribution of sports venues using reclassified POI (Point of Information) data, visualizes the distribution of intra-city population using population raster data from the WorldPop database, and analyses the distribution of sports venues and the urban population in Xi'an and Zhengzhou cities in 2020 from both the general and local perspectives based on various regression methods, such as MGWR, GWR, and linear fitting. The results show that the distribution of sports venues in Xi'an and Zhengzhou cities in 2020 was a good indicator of the population. The spatial distribution of sports venues and the population within the cities have a centre-periphery structure. From the global perspective, the distribution of sports venues is positively correlated with the intra-city population, and the promoting effect is significant. From the local perspective, the effect is spatially heterogeneous. Finally, this study explores the rationality of the complex impact and indicates that the research methodology can provide a reliable reference for other cities.

## 1. Introduction

Since the new era, with the continuous development of new urbanization, China's economy has rapidly improved, national income has increased, and residents' concerns have gradually shifted from growing material and cultural needs to the desire for a better life [1]. The intra-urban population refers to the population in the built-up areas of cities that are closely related to urban activities [2]. In 2014, "fitness for all" was officially promoted as a national strategy, and China's sports industry has ushered in the era of "sports for all", with sports and leisure activities becoming an important component of urban activities. As the core foundation and carrier of sports industry development, sports venues have assumed an irreplaceable role in

**Data Availability Statement:** The POI data for sports venues in Xi'an and Zhengzhou in the study were obtained from Baidu Map. The population data were obtained from WorldPop grid data and

the resident population data of the 7th National Population Census.

**Funding:** The author(s) received no specific funding for this work.

**Competing interests:** The authors have declared that no competing interests exist.

the structural reform of the sports supply side [3]. With the transformation of people's life needs, sports venues play an even more important role in society. Therefore, it is important to study the influence of sports venues on the population of urban built-up areas.

International scholars have conducted in-depth studies of sports venues and populations with rich results. In research on sports venues, infrastructure construction and the optimal operation of sports venues have been a long-standing and continuous focus of scholars [2, 4, 5]. The same is also true for the analysis of the operational interaction of sports equipment [6], utilization value [7], the operation mode and benefits [8], and other directions to promote the scientific use and replacement of sports equipment inside sports venues and the smart and low-carbon upgrade of service devices [9, 10]. Research also focuses on the role of sports venues in enhancing public health [11], socio-economic benefits [12] and the issue of sharing and opening school sports venues [13]. Research on the urban population is early and ongoing [14, 15], and the driving mechanisms [16] and effects [17] of population growth and decline are two major components that cannot be ignored. Higher urban productivity [18], innovation dynamism and output [19], a high-quality ecological environment [20], urban welfare [21], and diverse growth poles and public service provision [22] can drive population aggregation. At the same time, population is also an active factor in the spatial evolution of cities [23]. With the improvement in global world sportsmanship, the establishment of the International Day of Sport for Development and Peace, the implementation of China's national fitness, strong sports and Healthy China initiatives, and the United Nations call for active sports during the COVID-19 pandemic quarantine, the interest of the global public in physical activity is growing. As a vehicle for physical activity, studies have recognized the aggregation of population distribution [24], residents' well-being [25], personal physical activity levels [26], and regional quality of life [27]. The improvement in various aspects, such as the density of their distribution and the degree of their input, is increasingly linked.

First, there are relevant studies that point out that the influence of sports venues on populations is gradually becoming significant [28], and the arrival of the information age provides a new breakthrough for research in this area. Studies based on big data and new technologies are worth further exploration and experimentation. Second, most current research data on urban populations are based on administrative perspectives such as provinces and cities [28, 29], the population layout within cities urgently needs to be studied, and the impact of sports and leisure activities on the population distribution from a micro perspective is becoming increasingly obvious. Based on this research, the distribution of sports venues was characterized using reclassified POI data, and population raster data from the WorldPop database were used to visualize the distribution of the intra-city population.

As mega-cities with large populations, Xi'an and Zhengzhou are national central cities in the Yellow River Basin of China, and they attach great importance to the development of public sports. In this paper, we take Xi'an and Zhengzhou cities as examples to map the layout of sports venues and the inner-city population in 2020 based on multiple sources of data using kernel density analysis. Then, we use fitting analysis, OLS, GWR and MGWR to analyse the complex effects of sports venues on the layout of the inner-city population from the global and local dimensions (Fig 1) to improve the reliability and applicability of the results and to provide a scientific reference for the public in other cities. That is, the purpose of this study is to provide a scientific reference for the study of public sports facilities in other cities and to promote the sustainable development of cities.

The rest of this paper is organized as follows. Section 2 provides a general description of the study area. Section 3 describes the data sources and research methods of this paper. Section 4 presents the results of the study in blocks. Section 5 discusses the results of this study in the context of existing related studies. Section 6 summarizes the findings of this paper.

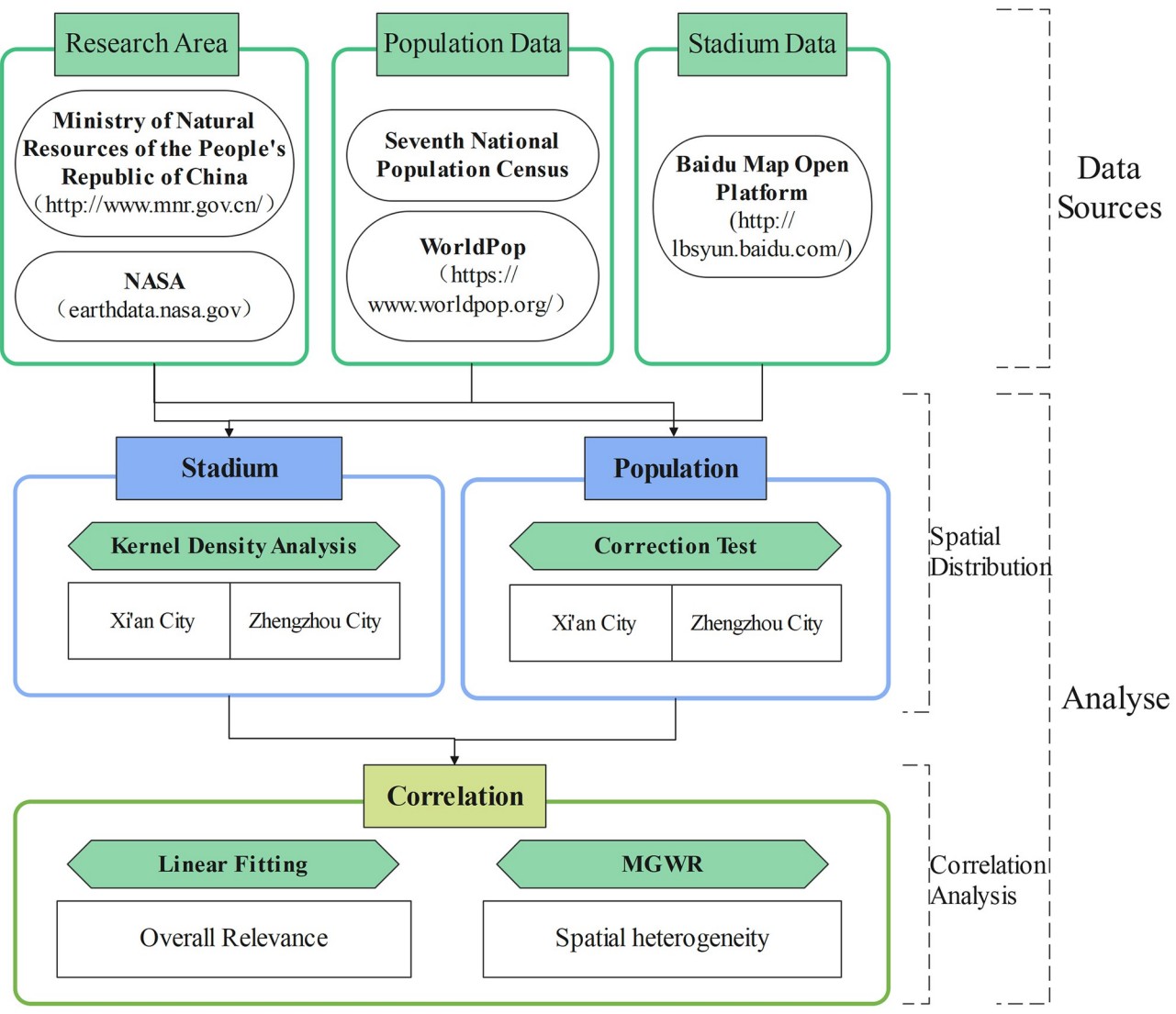

**Fig 1. Research approach.**

## 2. Overview of the study area

Xi'an city and Zhengzhou city are the capitals of Shaanxi Province and Henan Province, respectively, and are among China's 14 mega-cities. Xi'an has 11 districts and 2 counties, with a total area of 10,108 square kilometres, a built-up area of 868.2 square kilometres, and a resident population of 13,163,000 in 2021, making it an important central city in Western China according to the State Council. Zhengzhou city has 6 districts, 5 county-level cities and 1 county, with a total area of 7,567 square kilometres, a built-up area of 1,342.11 square kilometres and a resident population of 12,742,000 in 2021.

Xi'an city and Zhengzhou city are high-capacity national central cities that are clearly pointed out in the "13th Five-Year Plan for the Rise of Central Region" issued by the National Development and Reform Commission of China. They are also national central cities in the middle reaches of the Yellow River (Fig 2). They are modern metropoles that lead regional development and represent the national image. Taking Xi'an and Zhengzhou as examples to

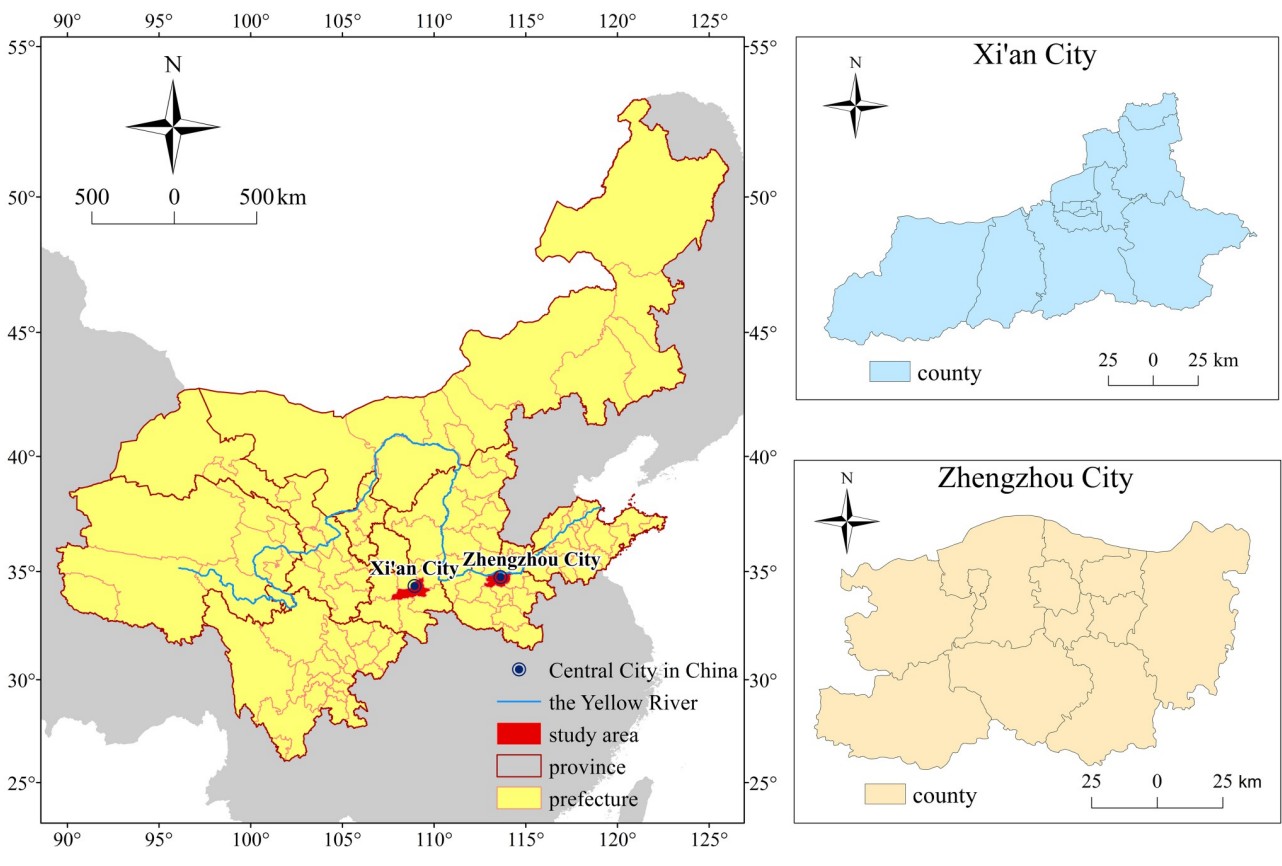

**Fig 2. Overview of the study area.**

study the relationship between sports venues and population has high representativeness and scientificity.

## 3. Materials and methods

### 3.1. Data sources

The study data consist mainly of stadium data and population data for 2020.

(1). The POI data for sports venues in Xi'an and Zhengzhou in the study were obtained from Baidu Map. These data include basic information such as the name, address, and geographic coordinates. After data cleaning, the number of sports venues in the two cities was obtained by combining the public information from the websites of the Xi'an and Zhengzhou Sports Bureaus and other websites: 12,316 and 14,972, respectively.

(2). The population data were obtained from WorldPop grid data and the resident population data of the 7th National Population Census. Census data are usually aggregated by administrative units on a cascading basis, and they are authoritative, systematic and standardized [30]. However, the administrative units on which such data rely do not coincide with the boundaries of the natural units in the actual study. These, these data are prone to the "variable element problem" [31] if the spatial distribution information of the population characterized by the average density of administrative districts cannot represent the spatial distribution characteristics of the population at a small scale. In contrast, WorldPop raster data downloaded

from the official website of WorldPop (https://hub.WorldPop.org/), with a spatial resolution of 100 m and high precision, can tap the spatial characteristics at the microscopic scale.

## 3.2. Data pre-processing

The study attempts to explore the impact of the distribution of sports venues on the inner-city population. Thus, the identification of the inner-city study area and the processing of open data are crucial.

**3.2.1. Determination of the inner-city study area.** The built-up area is the most densely populated area in a city, accounting for the largest proportion of the population. It is also a concentrated area for the distribution of sports venues. Taking the built-up area of cities as the final study area can improve the scientificity and reliability of the study. First, we convert the .tiff data from Landsat into vector data and extract the built-up area, build a 1 km*1 km fishing grid within the built-up area, and extract the part whose actual area is greater than or equal to one-half of the grid area to construct the inner-city study area (Fig 3).

**3.2.2. Stadium data.** The study uses Python 3.9 to crawl the POI data from the Baidu Open Platform 2020, performs data cleaning and data reclassification to extract the stadium data, and obtains 10,554 data points from Xi'an city and 10,035 data points from Zhengzhou city.

**3.2.3. Population raster data.** To better represent and explore the distribution pattern of the population, this study adopts the population raster data of WorldPop and uses county-level census data to correct and check the population data from the WorldPop database through partitioning so that the fitted value of each district is greater than 95% to ensure the rigor of the raster data. Finally, the population data were assigned to the fishing network within the study area based on the ArcGIS software platform to realize spatial visualization within the cities.

## 3.3. Research methodology

**3.3.1. Kernel density estimation.** Kernel density estimation (KDE) can estimate the probability density of the given sample data and is typically used to measure the density values of adjacent point-line elements to represent the spatial characteristics of the study object. Thus, it is widely used in spatial distribution research. In this paper, we use KDE to measure the spatial density of sports venues in Xi'an and Zhengzhou to obtain the spatial distribution maps of sports venues in Xi'an and Zhengzhou. The higher the value obtained is, the more clustered

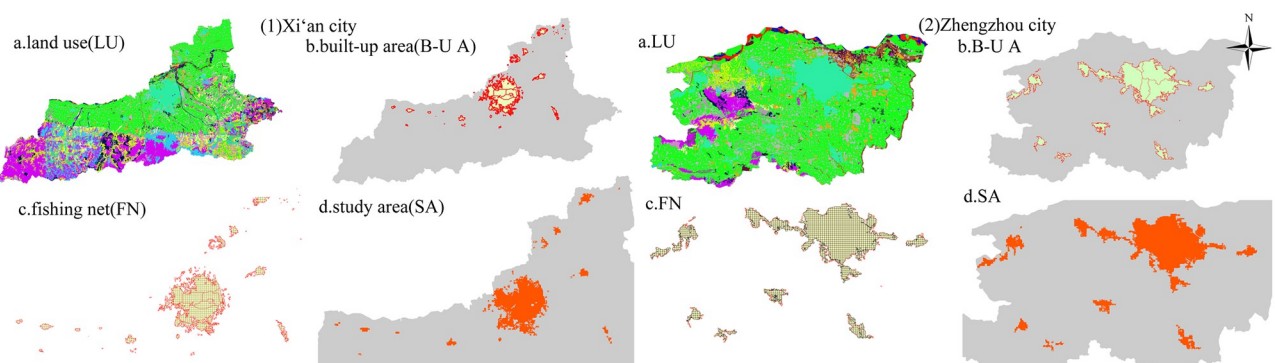

**Fig 3. Intra-urban study area: (a) land use; (b) extraction of the built-up area; (c) construction of the fishing network; (d) study area.**

the sports venues are in the area; in contrast, the lower the value obtained is, the more dispersed. The formula is shown as follows [32]:

$$\hat{f}(x) = \frac{1}{nh^2\pi} \sum_{i=1}^{n} \left[1 - \frac{(x-x_i)^2 + (y-y_i)^2}{h^2}\right]^2 \tag{1}$$

where $h$ is the threshold value and $n$ is the number of points in the threshold range and indicates the deviation from the interval.

In the calculation process, the search radius of the kernel density calculation is selected as the optimized default value, and the resulting image data have better smoothness and obvious local features. The default value is calculated as follows:

$$SearchRadius = 0.9*\min\left(SD, \sqrt{\frac{1}{\ln(2)}}*D_m\right)*n^{-0.2} \tag{2}$$

where $SD$ is the standard distance, $D_m$ is the median distance, the weight of each point is not set in the study of sports venues, and $n$ represents the number of points.

**3.3.2. Multi-scale geographically weighted regression (MGWR).** Geographically weighted regression (GWR) is a local regression model based on the construction of a spatial weight matrix. It takes into account the non-stationarity of space and is commonly used for spatial influence factor analysis. Multi-scale geographically weighted regression (MGWR) relaxes the assumption that all processes to be modelled are at the same spatial scale, which can solve the problem of smoothing the variables at their respective spatial levels, as well as the problem of using a specific bandwidth for each variable as an indicator of the role of each spatial process. The use of a multi-bandwidth approach can produce a spatial process model that is closer to reality and more useful. The formulation is expressed as follows [33]:

$$y_i = \sum_{j=i}^{k} \beta_{bjw}(u_i, v_i)x_{ij} + \varepsilon_i \tag{3}$$

where $y_i$ is the dependent variable of element $i$, $x_{ij}$ the attribute value of independent variable $j$ at location $i$, $\beta_{bjw}$ is the bandwidth used for the regression coefficient of the first variable $j$, ($u_i$, $v_i$) is the spatial coordinate of the first element $i$, and $\varepsilon_i$ is the residual.

$$\hat{\varepsilon} = y - \sum_{j=1}^{k} \hat{f}_j \tag{4}$$

MGWR sets the initial state with classical GWR and then calculates the initialized residual $\hat{\varepsilon}$, followed by an iterative calculation to find the optimal bandwidth.

In Eq (4), the initial residual $\hat{\varepsilon}$, additive term $\hat{f}_j$ and independent variable $x_i$ are GWR weighted on $i = 1\cdots n$ to find the optimum $\beta_{bjw}$. Then, the computation is continuously iterated until all variables $k$ are traversed. Finally, this cycle is iterated and repeated until the estimation converges to the convergence criterion.

## 4. Results

### 4.1. Spatial differentiation of sports venues based on POI data

The data on sports venues in Xi'an and Zhengzhou were used for density calculation by the kernel density analysis tool in ArcGIS10.2, the obtained density images were classified using the natural break method, and each of the two places was divided into five categories (Fig 4).

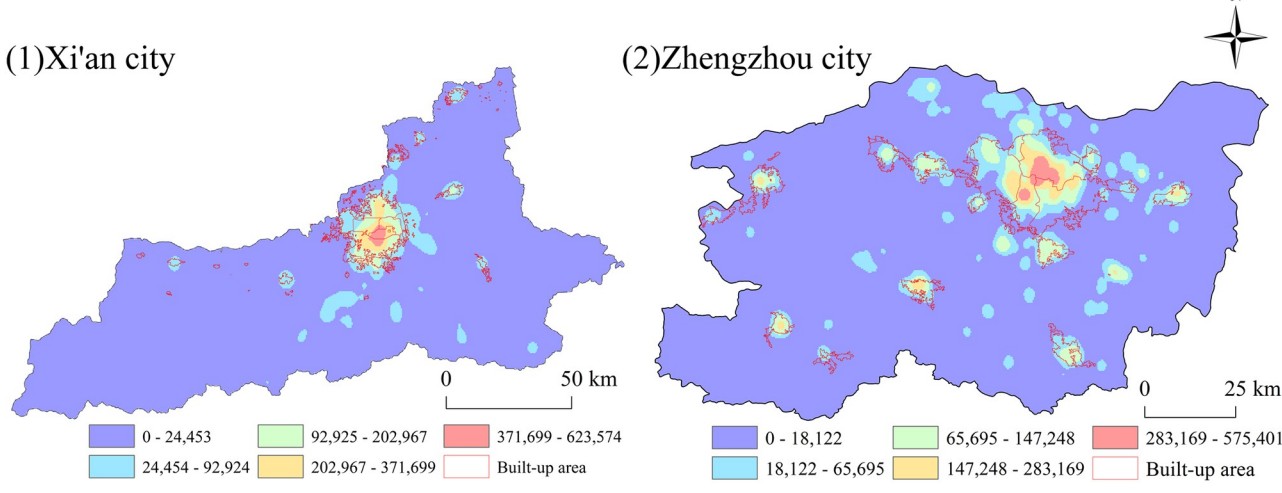

**Fig 4. Spatial distribution of sports venues (1: Xi'an city; 2: Zhengzhou city).**

Xi'an's sports venues show an obvious single-core centre-periphery concentric circular structure, with the highest value reaching 623,574. Additionally, the larger the value is, the more sports venues are distributed in the area. The centre is concentrated in the new city, Beilin District and Lianhu District, and continues to spread to the periphery. Zhouzhi County, the Eyi District and other peripheral districts and counties have a lower-density stadium distribution. Zhengzhou city sports venues form a clear multi-centre, multi-level distribution, with the highest value of density reaching 575,401. Higher density areas are concentrated in the Zhongyuan District, Jinshui District, Erqi District and Guanchenghuizi District, and the same Zhengzhou city district. The overall distribution corresponds to one main area (main urban area), one city (aviation city), three districts (western new urban area, southern new urban area and eastern new urban area) and four clusters (Gongyi, Dengfeng, Xinmi and Xinzheng) in the Zhengzhou city master plan (2010–2020).

Regarding the comprehensive distribution of sports venues in both Xi'an and Zhengzhou, we find that within the built-up area is the hot area where the sports venues are located. Urban construction land with basically complete municipal utilities within the built-up area of a city has convenient transportation, a high urbanization rate, and strong social vitality, and sports venues are densely distributed to promote the accessibility and convenience of mass sports and leisure activities.

## 4.2. Spatial distribution of the population within a city based on WorldPop data

To make the visualization results of the population distribution within a city scientific and reliable, the study rigorously corrected the population data based on the ArcGIS software platform and at the same time visualized only the population within the built-up area in a 1 km grid. Doing so made the spatial characteristics of the population distribution significantly clear (Fig 5).

As shown in Fig 5, the high-value area of the population in the inner city of Xi'an is similar to the distribution of its sports venues and shows a centre-periphery structure, mainly concentrated in the eastern part of the Lianhu District, the northern part of the Beilin District and the western part of the new city. Additionally, the whole high-value area coincides with the area

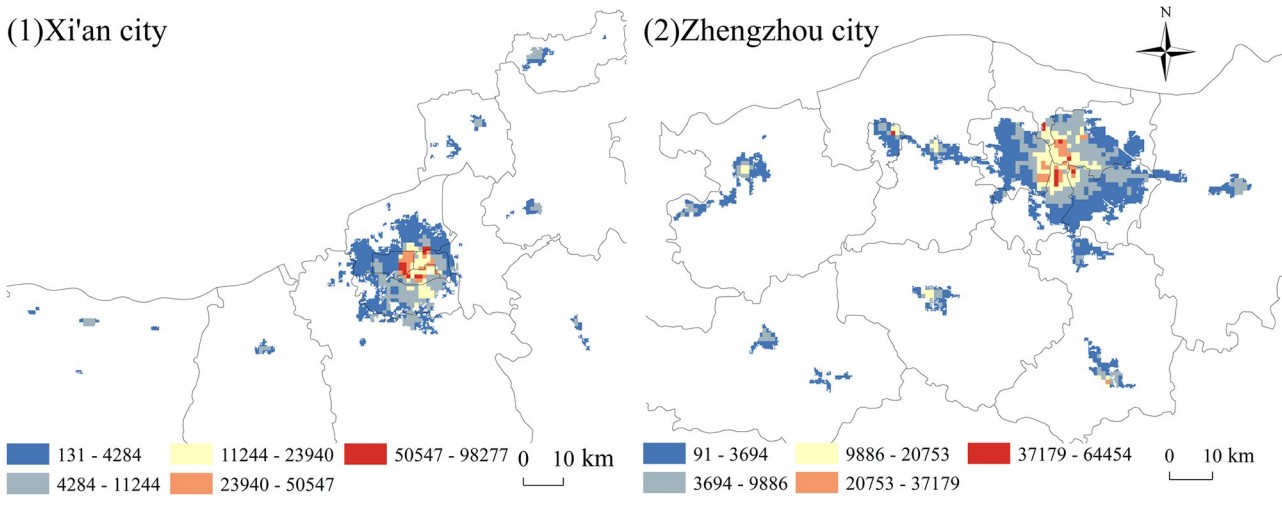

**Fig 5. Spatial distribution of the population within cities (1: Xi'an city; 2: Zhengzhou city).**

within the city walls of Xi'an. The area within the city walls is the central part of Xi'an, which is the earliest area in Xi'an to have the shape of a city. The area has well-developed transportation, with the Xi'an Railway Station, the Shaanxi Bus Station, several subways and several buses located within the city walls. It has a thriving economy and is home to several important commercial buildings.

The population within the city of Zhengzhou is concentrated in the Zhongyuan District, Jinshui District, Erqi District, and Guanchenghuizi District in a rhombus shape. All of this area is the main urban area of Zhengzhou, and it is worth noting that outside of the main urban area, the Shangdi District appears to be a high-value area for population distribution. The Shangjie District is not connected to the main urban area of Zhengzhou and is separated from Xingyang city. However, its core Shangjie village has rich aluminium resources and is the largest aluminium industrial base in China. Additionally, it is accompanied by the Shangjie Railway Station and is also the "throat of the Yellow River Journey". Thus, the population inside the city is large and concentrated.

### 4.3. Correlation analysis of sports venues and the intra-city population layout

**4.3.1. Overall correlation analysis.** Sports and leisure activities that enrich the spare time of the general public can represent people's aspiration for a better life and are closely related to the development of cities. Therefore, as a carrier of residents' leisure and sports activities, sports venues not only have the function of leisure and entertainment but also have certain social symbolic functions. Additionally, studies have proven that the distribution of sports venues is closely related to the size of the population [24, 28, 29]. To explore the influence of the density of the stadium distribution on the intra-city population distribution in Xi'an and Zhengzhou cities, scatter plots were created based on normalized data from Xi'an and Zhengzhou cities using Stata 16 and Origin 2018 software (Fig 6).

As shown in Fig 6, the models of Xi'an and Zhengzhou cities fit better based on the overall perspective, and the slopes of the fitted lines are positive. This means that the distribution of sports venues in both Xi'an and Zhengzhou cities can promote the growth of the intra-city population at the global scale. The slope of the fitted line is higher in Zhengzhou, which means

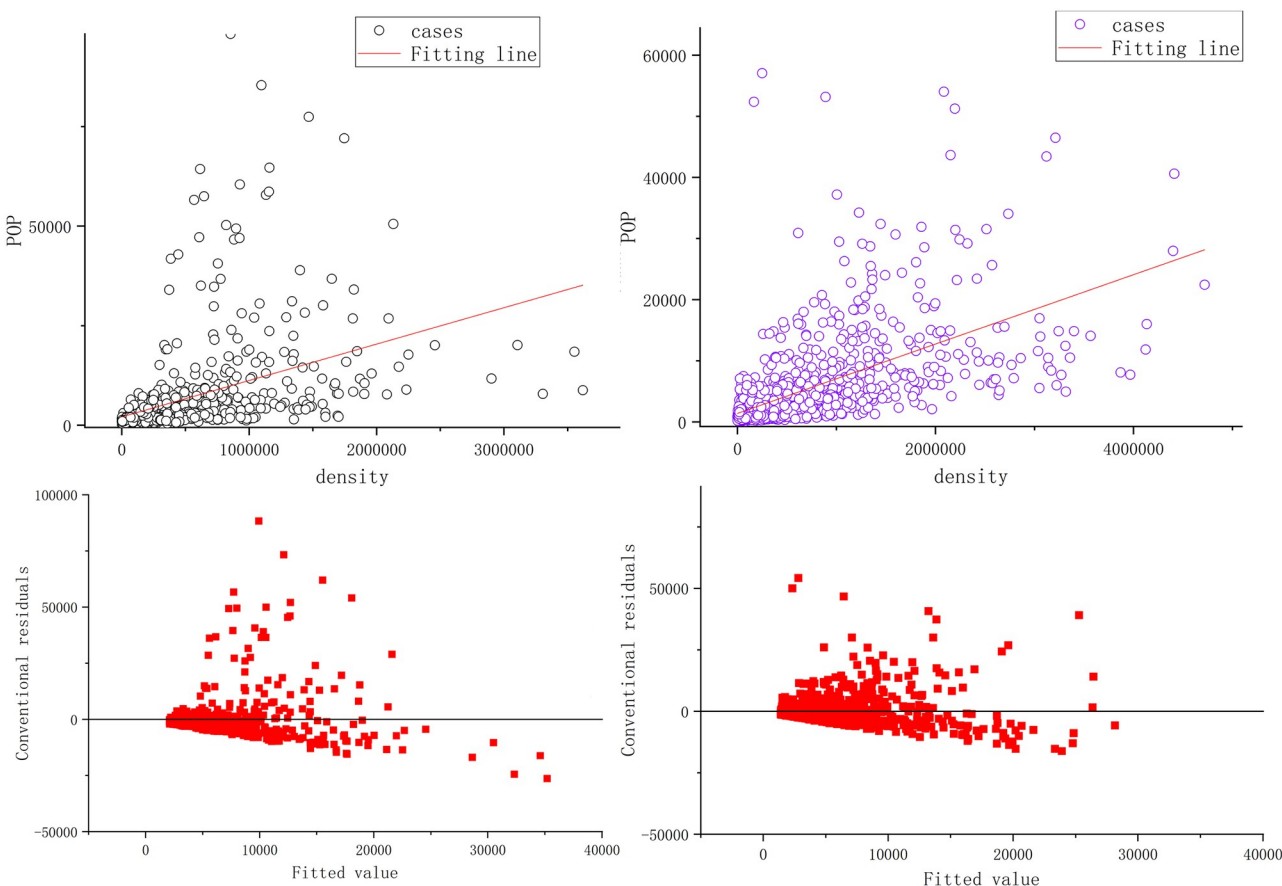

**Fig 6. Linear fit of the intensity of the sports stadium distribution and population density.**

that the intra-city population in Zhengzhou is more sensitive to the distribution of sports venues.

**4.3.2. Spatial heterogeneity analysis.** In this study, three regression analyses were attempted, namely, OLS, GWR, and MGWR. As shown in Table 1, the $R^2$ values of MGWR for both Xi'an and Zhengzhou cities were larger than those of the OLS and GWR models, at 0.68 and 0.71, respectively. Additionally, among the AICc values, for the value based on the MGWR model for Xi'an city, 1256.85, was much smaller than those based on the other two traditional models. The AICc values for Zhengzhou city were also smaller than those of the GWR and OLS models. The sum of the squared residuals of MGWR in Xi'an and Zhengzhou are also smaller than those of the traditional GWR and OLS models, indicating that the regression results of the MGWR model are more scientific and reliable, and the fit is better.

**Table 1. Parameter information for the OLS, GWR and MGWR regression fits.**

| Return method | Xi'an | | | Zhengzhou | | |
|---|---|---|---|---|---|---|
| | R2 | AICc | Residual Sum of Squares | R2 | AICc | Residual Sum of Squares |
| OLS | 0.18 | 1749.99 | 541.44 | 0.35 | 2703.15 | 731.53 |
| GWR | 0.53 | 1377.96 | 333.19 | 0.61 | 2204.81 | 423.44 |
| MGWR | 0.68 | 1256.85 | 210.39 | 0.71 | 2014.31 | 323.91 |

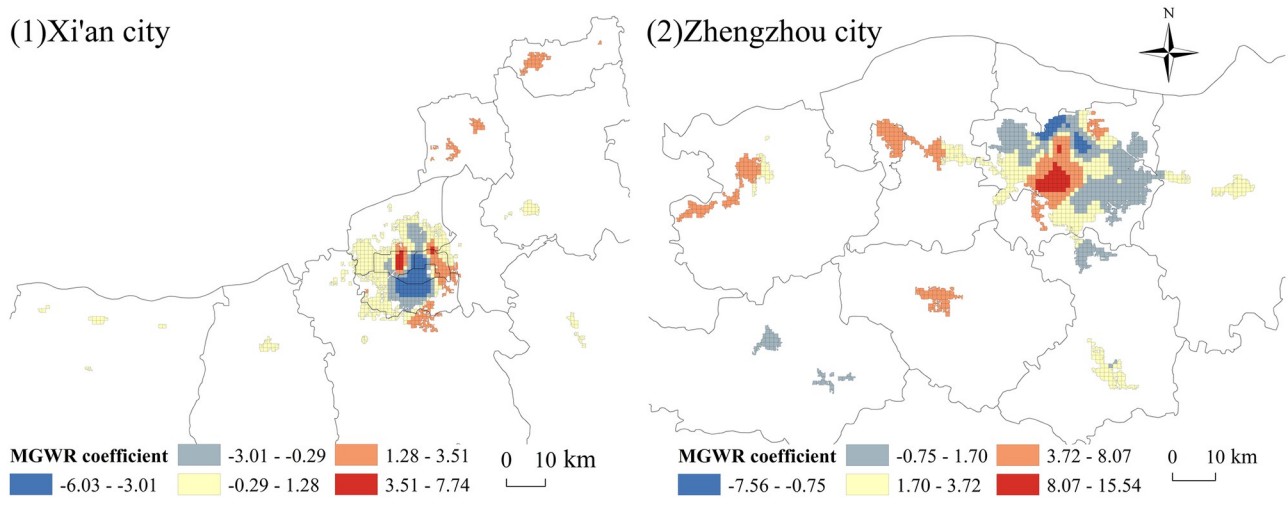

**Fig 7. Distribution of the MGWR regression coefficients for Xi'an and Zhengzhou.**

Combining the parameter ratios of the above models, the MGWR model is better than the traditional GWR model and OLS model in this study. Therefore, the MGWR model was selected for the study (Fig 7).

The distribution of the regression coefficients in Xi'an shows a centre-periphery structure. Additionally, the regression coefficient values from the centre to the periphery change from a negative to a positive correlation, showing an increasing trend. The negative correlation area expands from the Beilin District to the periphery, and it is distributed in the Lianhu District, Xincheng District, Yanta District, Weiyang District, and Lintong District. In this area, the distribution of sports venues resists the growth in the inner-city population, i.e., the more sports venues there are, the more the inner-city population tends to decrease. The Xi'an City Wall, which is the largest and best preserved ancient city wall in China, is a national 5A scenic spot, has tourist attractions such as the Xi'an Bell Tower, the Drum Tower, the Xi'an Bilin Museum, the Yongxing Square and other municipal facilities such as the Shaanxi provincial Government, and is the core area of the Xi'an CBD distribution, whose main functions are tourism, commerce and administration. The high density of stadium distribution occupies its main area. The high density of sports venues will take away the distribution of their main functions and therefore slow down the agglomeration of population. In the Beilin District and Yanta District, there are more higher education institutions (Xi'an Jiaotong University, Shaanxi Normal University, etc.), and there are better sports and recreation areas in the universities. Thus, the demand for public social sports venues is low. It is noteworthy that the central part of the Lianhu District and the eastern part of the new city have high positive correlation values. Both districts have similar properties, with the layout of old workers' family homes, low rents and a strong living atmosphere. The increase in sports venues optimizes the living environment of the district, improves the quality of life of district residents, and promotes an increase in the inner-city population.

Compared to Xi'an city, Zhengzhou city has the opposite distribution of regression coefficients, despite also having a centre-periphery structure. However, the regression coefficients change from a positive to a negative correlation from the centre to the periphery. The high-value area is located at the junction of the Zhongyuan District, Erqi District, Jinshui District and Guanchenghuizi District, and the distribution of sports sports venues in this area

contributes to population agglomeration. The area is distributed across numerous residential neighbourhoods and is also the location of the Zhengzhou Railway Station, which has an enormous number of migrant and permanent residents and a high demand for sports venues. As important support for public sports services, sports sports venues have a significant role in enhancing residents' well-being to residents' happiness. Additionally, sports sports venues have a positive effect on the intra-city population in areas with a high layout of settlements in the Shangdi District, Gongyi District and Xinmi City. The negative areas are located in the Zhongyuan District, Huizi District, Jinshui District, and Guancheng Huizu District. The Zhongyuan District is the gathering place of the Zhengzhou municipal government as well as other municipal units, and it is also the political and cultural centre of Zhengzhou. The low-value areas of the Huizi District, Jinshui District and Guancheng Huizuzizhi District are distributed with urban functional areas such as the Comprehensive Administrative Business Centre, Zhengzhou International Convention and Exhibition Centre, Yinji Trade City and Bairong World Trade Centre.

Overall, the distribution of sports venues in densely populated residential areas and transportation hub areas has a catalytic effect on the intra-city population, while the over-distribution of sports venues in CBDs, tourist attractions, and university gathering areas where special functions such as administration, business, tourism, and education are the main functions will slow down the increase in the intra-city population.

## 5. Discussion

### 5.1. Significance of research on urban built-up areas

The urban built-up area is the population agglomeration area within a city, and by the end of 2021, 559 million people in China were distributed within the built-up area. In this area, there are many human activities and strong social vitality. Moreover, the main contradiction of Chinese society is the contradiction between people's growing demand for a better life and unbalanced and insufficient development, and sports and leisure activities are one of the important factors in enhancing people's happiness in life. Therefore, studying the impact of the distribution of sports venues in built-up areas on the population holds great practical significance.

### 5.2. Complexity of the impact

From the results, it is clear that, in general, the increase in sports venues per unit area can attract an increase in population size. Synthesizing previous studies [34–36], sports venues, as the spatial carrier of diversified sports industries, are the foundation and pioneer industries for the development of sports industries. The construction and development of sports venues have a certain promoting effect on other sports industries. Sports venues provide venues for event operations and earn a certain amount of rent from them. This rent is used for daily maintenance and upgrades, and sports venues provide a place for consumers to watch games. Events provide content for consumers to watch games in the venue, and they also drive the development of related industries, such as sports training, sports brokerage, sports insurance, and sports advertising. These industries will ultimately provide corresponding services for event operations and obtain the corresponding service fees from event operations, laying the material foundation for the development of this industry. Event operations in sports lottery, sports goods, and sports tournaments provide opportunities for the derivation of industries such as sports lottery, sports goods, tourism, and sports games. A tournament will facilitate the advertising and promotion of brands and will obtain sponsorship from advertisers to maintain team management, tournament operations and tournament promotion. The construction of a stadium represents not just the development of a single industry; it will radiate to other sports

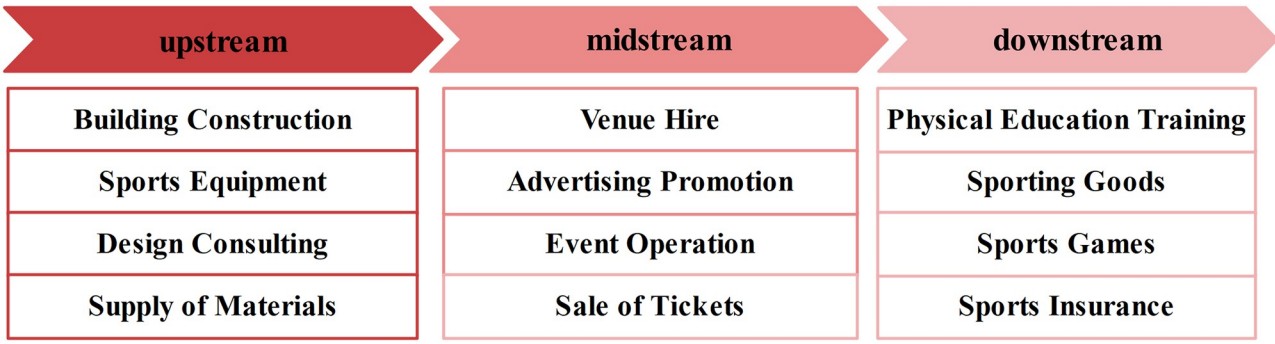

**Fig 8. Schematic flow of a simple industrial chain with sports venues as the core.**

industry sectors, forming a sports industry chain with the stadium as the core (Fig 8). This shows that sports venues play an important role in serving public sports, building the public sports service system and improving people's livelihood. Additionally, the scale and level of their development reflect the level of economic development of China's national and regional social civilization.

However, it is worth noting that the distribution of sports venues does not simply have a facilitating effect on the population, and their spatial heterogeneity is prominent. In areas where administrative, business, tourism and educational functions are the main functions, sports sports venues play an opposite role in population agglomeration. The over-distribution of sports venues increases the competition for land in the CBD, tourist attractions and other areas where they are located. Combined with "population push-pull theory" [37, 38], the distribution of sports venues enhances the well-being of residents and attracts an increase in the foreign population, thus forming a "population pull". At the same time, the distribution of sports venues causes a decrease in land resources and an increase in land rent, creating a "population pull". Within a region, the distribution of sports venues causes the "population push" to be greater than the "population pull", while in other regions, the "population pull" is greater than the "population push".

## 5.3. Advantages of the MGWR model

During the analysis, the study used three regression models, OLS, GWR, and MGWR. The OLS model, as a benchmark model, is simple and easy to operate and runs faster. The GWR model, as a spatial analysis technique, is more accurate because it takes into account the local effects of spatial objects in its calculation. The MGWR model, similar to the GWR model, is also a local regression model, but it is one in which the coefficient values can vary across space. Additionally, the bandwidth used to define the neighbourhood around each element can vary across the explanatory variables, which allows the model to capture different proportions of the relationship between the explanatory and dependent variables. Through comparative analysis, the MGWR model has higher scientific validity and better fits the needs of this study due to its multi-scale characteristics, which form the corresponding bandwidths based on different degrees of data point clustering. At the same time, it is found that the model fits better, so it is worth promoting and using the MGWR model in research related to the distribution of sports venues and the population distribution to obtain more realistic and reliable research results.

### 5.4. Limitations of the study

At the same time, there are still some shortcomings in this study. First, most of China's population is still distributed in areas outside built-up urban areas, and this paper does not explore the effect of sports venues on the population in such areas. Second, this study focuses on the impact only in 2020, and the mechanism of the impact can be explored in the future by expanding the time series of the study and considering the joint effect of other control variables, such as the economic, ecological, and industrial structures, on the intra-city population. This study conducts a preliminary exploration of the impact of the stadium distribution on the intra-city population. Therefore, this research paradigm can be extended to relevant studies in other regions to obtain a larger range of empirical cases and then make more in-depth progress in studies related to the distribution of sports venues to ensure scientific and reliable urban planning.

## 6. Conclusions

Based on the POI data of sports sports venues in Xi'an and Zhengzhou cities and WorldPop population data, this study uses FDE and MGWR to explore the impact of the sports stadium distribution on the intra-city population in Xi'an and Zhengzhou cities, and the following conclusions are obtained.

(1). The spatial distribution of sports sports venues and the intra-city population in Xi'an and Zhengzhou have a "centre-periphery" structure. Additionally, both sports sports venues and the intra-city population have a high density of distribution within built-up areas, but the distribution of each city has its own characteristics. Xi'an has a single centre-periphery structure with concentric circles, with the Xincheng District, Beilin District and Lianhu District as the centres, spreading to the peripheral districts and counties, and gradually decreasing in density, while Zhengzhou has formed an obvious multi-centre, multi-level structure. The high-value area of Xi'an is related to the high-value area of the city. The high-value population area in Xi'an city coincides with the area within the city walls of Xi'an. The high-value area in Zhengzhou city has a rhombus shape, mainly concentrated in the Zhongyuan District, Jinshui District, Erqi District, and Guanchenghuizi District.

(2). From the global perspective, the distribution of sports venues and the intra-city population in both Xi'an and Zhengzhou have a positive correlation, the model fits well in both places, and the slope of the fitted line is positive in both places. The slope of the fitted line is higher in Zhengzhou city, which indicates that the intra-city population in Zhengzhou city is more sensitive to the distribution of sports venues.

(3). The distribution of sports venues in Xi'an and Zhengzhou is spatially heterogeneous in terms of its effect on the intra-city population, and the range of its contribution is relatively large. The areas where sports venues have a deterrent effect on population clustering within the built-up area of Xi'an are mainly located within the city walls of Xi'an (Beilin District, Lianhu District and Xincheng District), while they are scattered in the Yanta District, Weiyang District and Lintong District. The areas where the distribution of sports venues in Zhengzhou has a negative effect on the population are located in the Zhongyuan District, Huizi District, Jinshui District and Guancheng Huizu District. The analysis found that the distribution of sports sports venues makes a significant contribution to the population size within a city, except for areas with special functions such as administration, business, tourism and education.

Therefore, it is recommended that in the planning of sports sports venues, on the basis of fully considering the relationship between the distribution of sports sports venues and the inner-city population in a certain area, great attention should be paid to the spatial

heterogeneity of the two, and more sports sports venues should be planned appropriately due to the need in the area of the positive correlation between the two to maximize the role of sports sports venues in inner-city population agglomeration, driving population growth in the relevant areas and driving regional development.

## Acknowledgments

We express our sincere thanks to the editor and anonymous reviewers for their comments and suggestions, which considerably helped to improve the quality of this study. We are solely responsible for the opinions expressed in this study.

## Author Contributions

**Conceptualization:** Shulin Zhang, Yang Liu.

**Data curation:** Xuejie Zhang.

**Formal analysis:** Shulin Zhang, Yang Liu.

**Methodology:** Shulin Zhang.

**Project administration:** Yang Liu.

**Resources:** Xuejie Zhang.

**Software:** Xuejie Zhang.

**Supervision:** Yang Liu.

**Validation:** Xuejie Zhang.

**Visualization:** Xuejie Zhang.

**Writing – original draft:** Shulin Zhang.

**Writing – review & editing:** Shulin Zhang, Xuejie Zhang.

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
