## [Decision Letter · Decision Letter 0]

14 Feb 2023

PONE-D-22-35291A Study of the Influence of Sports venues on the Intra-City Population Layout based on Multi-Source Data—Taking Xi'an City and Zhengzhou City as ExamplesPLOS ONE

Dear Dr. Liu,

Thank you for submitting your manuscript to PLOS ONE. After careful consideration, we feel that it has merit but does not fully meet PLOS ONE’s publication criteria as it currently stands. Therefore, we invite you to submit a revised version of the manuscript that addresses the points raised during the review process.

We look forward to receiving your revised manuscript.

Kind regards,

Jing Cheng

Academic Editor

PLOS ONE

Journal Requirements:

4. PLOS requires an ORCID iD for the corresponding author in Editorial Manager on papers submitted after December 6th, 2016. Please ensure that you have an ORCID iD and that it is validated in Editorial Manager. To do this, go to ‘Update my Information’ (in the upper left-hand corner of the main menu), and click on the Fetch/Validate link next to the ORCID field. This will take you to the ORCID site and allow you to create a new iD or authenticate a pre-existing iD in Editorial Manager. Please see the following video for instructions on linking an ORCID iD to your Editorial Manager account: https://www.youtube.com/watch?v=_xcclfuvtxQ.

5. We note that Figures 2-5 and 7 in your submission contain [map/satellite] images which may be copyrighted. All PLOS content is published under the Creative Commons Attribution License (CC BY 4.0), which means that the manuscript, images, and Supporting Information files will be freely available online, and any third party is permitted to access, download, copy, distribute, and use these materials in any way, even commercially, with proper attribution. For these reasons, we cannot publish previously copyrighted maps or satellite images created using proprietary data, such as Google software (Google Maps, Street View, and Earth). For more information, see our copyright guidelines: http://journals.plos.org/plosone/s/licenses-and-copyright.

a) You may seek permission from the original copyright holder of Figures 2-5 and 7 to publish the content specifically under the CC BY 4.0 license.  

Natural Earth (public domain): http://www.naturalearthdata.com/.

Reviewers' comments:

Reviewer's Responses to Questions

**Comments to the Author**

1. Is the manuscript technically sound, and do the data support the conclusions?

Reviewer #1: Yes

Reviewer #2: Yes

2. Has the statistical analysis been performed appropriately and rigorously? 

Reviewer #1: Yes

Reviewer #2: Yes

3. Have the authors made all data underlying the findings in their manuscript fully available?

Reviewer #1: Yes

Reviewer #2: Yes

4. Is the manuscript presented in an intelligible fashion and written in standard English?

Reviewer #1: Yes

Reviewer #2: Yes

5. Review Comments to the Author

Reviewer #1: This study investigates the influence of sports venues on the Population Layout in two mega cities in china in the year 2020. The paper is neat and well-written and the study is interesting and scientifically sound. However, some minor issues need some attention, including:

1. Lines 9, 300, 301, 354, 395, 396, 419, 422,423, 425, and 427: Please delete the duplicated word of "sports". Otherwise, the authors need to point out clearly what this duplication stands for?!

2. Lines 20-21: "Finally, ... cities". This statement is confusing and NOT clear and better if being rephrased in a straightforwardly meaning.

3. Line 65: "Based on this research". Please clarify (and later rephrase), "this research". refers to the current study, or the cited previous study (28 & 29)?!

4. Line 113: Please delete the duplicated "these".

5. Line 146: "3.3. Research Methodology": Using the terminology of "Research Methodology" is extremely confusing here, especially as a subtitle under the section of "Material and Methods". Alternatively, I strongly suggest using "Statistical Method", instead.

6. Line 345: "This shows". "This" What? Please specify.

7. Line 378: "5.4. Limitations of the Study". This section is to be placed after the Conclusion Section. And please delete "of the Study".

8. Line 379: "At the same time, ...study". Please delete "at the same time".

9. Lines 395, 406, and 411; Please delete the paragraph numbering i.e. (1, 2, and 3).

Reviewer #2: The manuscript is well written and provides a sound understanding of the set objective it outlines in the introduction. The study examined the influence of the distribution of sport venues on the intra-city population layout in Xi’an city and Zhengzhou city. Based on spatial analysis using MGWR, GWR and linear fitting, the study found that the spatial distribution of sport facilities venue is a good indicator of population distribution. While the manuscript is good, it can be further improved and its chances of being published greatly increased if the comments made in my review are carefully considered and the manuscript duly revised.

6. PLOS authors have the option to publish the peer review history of their article (what does this mean?). If published, this will include your full peer review and any attached files.

Reviewer #1: No

Reviewer #2: **Yes: **Regina Obilie Amoako-Sakyi

---

## [Author Response · Author response to Decision Letter 0]

28 Mar 2023

Response to Editor Comments

Thanks to the editor's reminder, we added a description of the map to the original manuscript.

Response to Reviewer 1 Comments

Dear PLOS ONE Editorial Board Editors and Reviewers.

Hello!

Thank you very much for your review of "Influence of sports venues on the intra-city population layout based on multi-source data. A case of Xi'an city and Zhengzhou city". We have addressed the reviewer's questions. We have carefully answered the reviewer's questions and made detailed revisions to the paper according to the reviewer's comments, and the revised parts are marked in red with revision status. The answers to the questions raised by the reviewers are as follows.

Point 1: Lines 9, 300, 301, 354, 395, 396, 419, 422,423, 425, and 427: Please delete the duplicated word of "sports". Otherwise, the authors need to point out clearly what this duplication stands for?!

Response 1: Thank you for your suggestion. In response to your suggestion, we checked and found that the duplicate word "sports" appeared in lines 9, 300, 301, 354, 395, 396, 419, 422, 423, 425, 427 and have now deleted the duplicate word.

Point 2: Lines 20-21: "Finally, ... cities". This statement is confusing and NOT clear and better if being rephrased in a straightforwardly meaning.

Response 2: The phrase "Finally, ... cities" has been replaced with "The study emphasizes the need to scientifically plan the layout of sports venues in a targeted way to reasonably promote the sustainable and high-quality development of different regions." in response to your suggestion.

Point 3: Line 65: "Based on this research". Please clarify (and later rephrase), "this research". refers to the current study, or the cited previous study (28 & 29)?!

Response 3: The phrase "Based on this research" has been changed to "Based on current research" to refer to the current study.

Point 4: Line 113: Please delete the duplicated "these".

Response 4: Thank you for your question. After checking, we found a duplicate "these" in line 113 and have removed the duplicate word.

Point 5: Line 146: "3.3. Research Methodology": Using the terminology of "Research Methodology" is extremely confusing here, especially as a subtitle under the section of "Material and Methods". Alternatively, I strongly suggest using "Statistical Method", instead.

Response 5: Thank you for your suggestion. After our discussion and combining your suggestions, we have changed the title of 3.3.

Point 6: Line 345: "This shows". "This" What? Please specify.

Response 6: Thank you for your question. "This shows" in line 345 is expressed through the industry chain.

Point 7: Line 378: "5.4. Limitations of the Study". This section is to be placed after the Conclusion Section. And please delete "of the Study".

Response 7: In response to your suggestion, we have placed line 378, "5.4. Limitations of the Study" after the conclusion section, and deleted "of the Study" and replaced it with "7. Limitations "

Point 8: Line 379: "At the same time, ...study". Please delete "at the same time".

Response 8: In response to your suggestion, "At the same time, . . study" by deleting "at the same time".

Point 9: Lines 395, 406, and 411; Please delete the paragraph numbering i.e. (1, 2, and 3).

Response 9: Paragraph numbers (1, 2, 3) on lines 395, 406 and 411 have been removed as per the suggestion you gave.

Yours sincerely,

Yang Liu

Response to Reviewer 2 Comments

Dear PLOS ONE Editorial Board Editors and Reviewers.

Hello!

Thank you very much for your review of "Influence of sports venues on the intra-city population layout based on multi-source data. A case of Xi'an city and Zhengzhou city". We have addressed the reviewer's questions. We have carefully answered the reviewer's questions and made detailed revisions to the paper according to the reviewer's comments, and the revised parts are marked in red with revision status. The answers to the questions raised by the reviewers are as follows.

i. Topic

Point 1: I suggest the topic is modified to read: “Influence of sports venues on the intra-city population layout based on multi-source data. A case of Xi’an city and Zhengzhou city”

Response 1: Thank you for your suggestions, we have revised the article in response to your suggestions.

ii. Abstract

Point 1: It will be helpful if the abstract includes a couple of recommendations based on the findings of the study as stated in the abstract section. Furthermore line 9 should be checked for a repetition of the word ‘sports’.

Response 1: Thank you for your careful reading of the study abstract, which begins with the last sentence of the abstract presenting: The study recommends that the layout of sports venues should be planned in a targeted and scientific manner, controlling the increase of sports venues in areas with special functions such as administration, business, tourism and education, while planning more sports venues in other areas appropriately due to needs, and reasonably promoting the sustainable and high-quality development of different areas; After checking the ninth line there is a duplicate of "sports", and the duplicate word was deleted.

iii. Introduction

Point 1: The introduction is well written.

Response 1: Thank you for your recognition and compliments!

iv. Materials and methods

It will be good if authors could respond to the following issues and make them clearer in the materials and methods section of the manuscript: 

Point 1: Kindly clarify the paragraph (line 104 to 108) in terms of the two sets of data used for the study. Why was there a need for the second dataset (data from the Public Information website)

Response 1: Thank you for your question, first of all, POI data is a kind of emerging open data, in order to improve the scientific nature of the study, we used the more reliable Sports Authority website to verify the data; at the same time, we re-examined the original manuscript and deleted the inappropriate words, and now the paragraph is as follows: The POI data for sports venues in Xi'an and Zhengzhou in the study were obtained from Baidu Map. These data include basic information such as the name, address, and geographic coordinates. After data cleaning, the number of sports venues in the two cities was obtained by combining the public information from the websites of the Xi'an and Zhengzhou Sports Bureaus: 12,316 and 14,972, respectively.

Point 2: There should be consistency in the use of ‘inner city’ or ‘intra-city’ which seems to be used interchangeably in the manuscript.

Response 2: We have carefully read the original manuscript and standardized its language, using the term "intra-city".

Point 3: Line 128, how do you convert .tiff data to vector and extract the built-up area? Was there no image classification done? An explanation of the process will do.

Response 3: The process of converting .tiff data into vector data is tedious and not the focus of this paper, so the process is explained as follows: using Landsat TM/ETM/OLI remote sensing images as the main data source, after image fusion, geometric correction, image enhancement and stitching, the national land use types are classified into 6 primary classes, 25 secondary classes and some tertiary classes by human-computer interaction visual interpretation. The land use data are classified into 6 primary categories, 25 secondary categories and some tertiary categories.

Point 4: At line 146, I would suggest you use the caption: data analysis.

Response 4: We have changed "Research Methodology" to "Data Analysis" in response to your suggestion.

Point 5: Which analytical software was used in this study?

Response 5: In this study, ENVI5.3 and ArcGIS10.6 were used in the data pre-processing stage, ArcGIS10.6 was used for the kernel density estimation method, and MGWR2.2 was used for the multi-scale geo-weighted regression.

Point 6: In the introduction and abstract, you mention the use of OLS and Linear fitting as part of the analytical tool used in your study, however, it has not been explained in your data analysis section.

Response 6: In this study, OLS, GWR and MGWR were used in doing regression analysis to obtain their regression parameters R2, AICc and residual sum of squares respectively and to compare the obtained parameters, it was concluded that MGWR was more suitable for this study, so OLS, GWR and MGWR were used, while only the results of MGWR were needed in the analysis part.

v. Results

Point 1: Unfortunately, authors failed to mention that they run correlation analysis in their write up at the data analysis section even though this is mentioned and discussed at the results section. 

Response 1: Thank you for raising this issue and we have added a correlation analysis to the data analysis section as follows:

3.3.2. Pearson Correlation Analysis

Pearson correlation refers to the simple linear correlation between random variables X and Y whose joint distribution obeys a two-dimensional normal distribution. The correlation between X and Y is represented by the simple correlation coefficient r. The Pearson correlation coefficient is used to measure whether the two data sets are on a straight line and is calculated as follows [33]:

(5)

The value of Pearson correlation coefficient ranges from [-1, 1], when r=0, it means there is no linear relationship between the two; when r>0, there is a certain linear positive relationship between the two, which means X has a certain positive relationship to Y, and the opposite is negative.

Point 2: Again, the scatter plots show a normal relationship than a correlation analysis. If it was a correlation analysis, kindly present the significance level.

Response 2: To represent the results of correlation analysis more clearly, we added the results of Pearson correlation analysis in 4.3.1 and changed Figure 6 to add the correlation coefficient heat map.

4.3.1. Overall Correlation Analysis

Sports and leisure activities that enrich the spare time of the general public can represent people's aspiration for a better life and are closely related to the development of cities. Therefore, as a carrier of residents' leisure and sports activities, sports venues not only have the function of leisure and entertainment but also have certain social symbolic functions. Additionally, studies have proven that the distribution of sports venues is closely related to the size of the population [24,28,29]. To explore the influence of the density of the stadium distribution on the intra-city population distribution in Xi'an and Zhengzhou cities, scatter plots and Pearson correlation analysis were created based on normalized data from Xi'an and Zhengzhou cities using Stata 16 and Origin 2018 software (Figure 6).

Figure 6.Heat map of linear fit and correlation coefficient between the intensity of sports stadium distribution and population density.

As shown in Figure 6, the models of Xi'an and Zhengzhou cities fit better based on the overall perspective, and the slopes of the fitted lines are positive. The coefficients obtained from the Pearson correlation analysis were all positive and significant at the 0.001 level. This means that the distribution of sports venues in both Xi'an and Zhengzhou cities can promote the growth of the intra-city population at the global scale. The slope of the fitted line and the correlation coefficient are higher in Zhengzhou, which means that the intra-city population in Zhengzhou is more sensitive to the distribution of sports venues.

Point 3: It will be useful to explain the OLS results in the data analysis section as presented in Table. 

Response 3: After the comparison of the three regression methods, it was found that the r2 of OLS was extremely small and the value of AICc was too large, and the regression effect was not satisfactory. The authors, after all the discussions, concluded that the results obtained from the OLS regression model could not better explain the influence of stadium distribution on population layout.

Yours sincerely,

Yang Liu

---

## [Decision Letter · Decision Letter 1]

27 Apr 2023

Influence of sports venues on the intra-city population layout based on multi-source data: A case of Xi'an city and Zhengzhou city

PONE-D-22-35291R1

Dear Dr. Liu,

We’re pleased to inform you that your manuscript has been judged scientifically suitable for publication and will be formally accepted for publication once it meets all outstanding technical requirements.

Kind regards,

Jing Cheng

Academic Editor

PLOS ONE

Additional Editor Comments (optional):

Reviewers' comments:

Reviewer's Responses to Questions

**Comments to the Author**

1. If the authors have adequately addressed your comments raised in a previous round of review and you feel that this manuscript is now acceptable for publication, you may indicate that here to bypass the “Comments to the Author” section, enter your conflict of interest statement in the “Confidential to Editor” section, and submit your "Accept" recommendation.

Reviewer #1: All comments have been addressed

Reviewer #2: All comments have been addressed

2. Is the manuscript technically sound, and do the data support the conclusions?

Reviewer #1: Yes

Reviewer #2: Yes

3. Has the statistical analysis been performed appropriately and rigorously? 

Reviewer #1: Yes

Reviewer #2: Yes

4. Have the authors made all data underlying the findings in their manuscript fully available?

Reviewer #1: Yes

Reviewer #2: Yes

5. Is the manuscript presented in an intelligible fashion and written in standard English?

Reviewer #1: Yes

Reviewer #2: Yes

6. Review Comments to the Author

Reviewer #1: All comments have been addressed in the revised version of the manuscript. There is no more comment(s) to be addressed.

Reviewer #2: (No Response)

7. PLOS authors have the option to publish the peer review history of their article (what does this mean?). If published, this will include your full peer review and any attached files.

Reviewer #1: No

Reviewer #2: **Yes: **Regina Obilie Amoako-Sakyi

---

## [Editor Report · Acceptance letter]

2 May 2023

PONE-D-22-35291R1 

A Study of the Influence of Sports venues on the Intra-City Population Layout based on Multi-Source Data—Taking Xi'an City and Zhengzhou City as Examples 

Dear Dr. Liu:

I'm pleased to inform you that your manuscript has been deemed suitable for publication in PLOS ONE. Congratulations! Your manuscript is now with our production department. 

Kind regards, 

on behalf of

Dr. Jing Cheng 

Academic Editor

PLOS ONE